# Spatial Heterogeneity and Coupling of Economy and Population Gravity Centres in the Hengduan Mountains

**Yong Luo [1], Hui Yu [2,\*], Siyuan Liu [1], Yuting Liang [1] and Shaoquan Liu [2]**

[1]  College of Earth Sciences, Chengdu University of Technology, Chengdu 610059, China;
    luoyong2014@cdut.edu.cn (Y.L.); liusiyuan7573@sina.com (S.L.); liangyuting64@sina.com (Y.L.)

[2]  Institute of Mountain Hazards and Environment, Chinese Academy of Sciences, Chengdu 610041, China;
    liushq@imde.ac.cn

\*  Correspondence: yuhui@imde.ac.cn

**Abstract:** The junction region of the Sichuan–Yunnan–Guizhou province (JRSYG) is a multinational community that includes the Hengduan mountainous areas, although its economy is lagging behind other regions in China. This study explored the spatial heterogeneity and coupling of economic and population gravity centres during the period 1995–2015, and determined the hotspots driving the economic development in the JRSYG region. We obtained the following results: (1) The global Moran's *I* and the Moran scatter plots of the economy and population showed that the distribution of economy and population was a nature clustering. The scatter plots of the economy and population were mainly distributed in Quadrant III, with an L–L gathering. With the passage of time, the agglomeration and driving effects of the economy become stronger. (2) By the dynamic evolution process of the economy and population, the economy gravity centre (GE) and population gravity centre (GP) were approaching each other during the period 1995–2015. The equilibrium points showed an inverted U-shaped curve for the past few years. The spatial coupling of GE and GP increased every year. The balanced degree of regional development continuously improved. (3) The economic development level showed a polarisation pattern with a southwest growth pole and northeast growth pole. The point–axis spatial development pattern is presented, with two economic hotspots (Panzhihua and Luzhou) and three sub-hotspots (Xichang, Zhong shan, Zhaoyang). If further advantages from policy and infrastructure support are obtained, the hot poles can drive the social and economic development of the surrounding regions, which will alleviate regional differences in the future.

**Keywords:** economy gravity centre; population gravity centre; coupling state; spatial heterogeneity; point–axis spatial development pattern

## 1. Introduction

China has the world's second-largest economy [1], with politicians and regional scientists having placed great importance on the issue of regional economic inequality [2]. The economy and population gravity centre are key indicators for measuring migration and the driving forces of regional change. They have recently become hot topics of interest for domestic and foreign researchers [3–8], and their research outcomes have had a profound impact on the population and the economy [9–11]. Their main focus has been on the national [12–15] and provincial levels [16–19].

Economic concentration did not produce a corresponding process of population concentration within the same period, which is the main reason that there is a significant regional development gap in China [13]. Hence, the spatial convergence and heterogeneity of regional economy growth has

become a new subject in economic and human geography [20–24]. However, China's current research on regional economic differences has focused on developed regions [25], large-scale regions [26], and multiscale regions, such as the eastern, middle, and western regions of China [27–30]. However, mountainous areas have rarely been studied, because they are located far from provincial centres and administrative boundaries limit the diffusion of economic activities [31]. Along with the *Western Development Strategy* and the *Yangtze River Economic Belt Development Strategy* proposed in recent years [32], it is necessary to further study mountainous areas and interprovincial boundary regions.

Compared with developed regions, the junction region of the Sichuan–Yunnan–Guizhou province (JRSYG) is a multinational community. In this community, the overall level of social and economic development is lagging behind. This is mainly in the mountainous areas of each province, which are located far away from the political, economic and cultural centres. However, this forms the central region of southwest china, and together constitutes the core of the southwest. There is insufficient research on the concentrated and contiguous poor mountainous areas.

Exploratory spatial data analysis (ESDA), the gravity centre model, and the spatial coupling model were used to measure and estimate the coupling degree and spatial heterogeneity of the economy and population gravity centres during the period 1995–2015 in the JRSYG. These efforts aimed to further explore the causes of change and their relationship to the process of spatial equilibrium in regional development. It has become a main trend that China's economic gravity centre is moving from inland to the coastal areas, while China's population gravity centre has shifted to the southeast [13]. Thus, it is important to find the local economic hotspots to attract the western population to remain in their original areas for employment. If further advantages from policy and infrastructure support are obtained, the growth poles in the JRSYG can promote provincial cooperation and increase the competitiveness of the west. The specific aims of this study were: (1) to determine the spatial heterogeneity of the economy and population in the JRSYG; (2) to illustrate the spatio-temporal development and coupling of economic and population gravity centres in the JRSYG over the past 20 years, by determining the hot poles that can drive the social and economic development of the surrounding regions. This will allow us to enhance the competitiveness of the west and reduce the local population moving to the eastern coastal areas, which will promote regional sustainable development. (3) Finally, we want to provide reliable policy recommendations for the balanced and reasonable distribution of population and economic resources in the JRSYG.

## 2. Materials and Methods

### 2.1. Study Area

The JRSYG, which has a complex and diverse natural environment and an extremely fragile ecological environment, is located in the first stage of the Qinghai–Tibet plateau step ladder in the second level transition zone. This area involves 5 cities and 40 counties in Sichuan, 5 cities and 20 counties in Yunnan, and 3 cities and 14 counties in Guizhou (Figure 1).

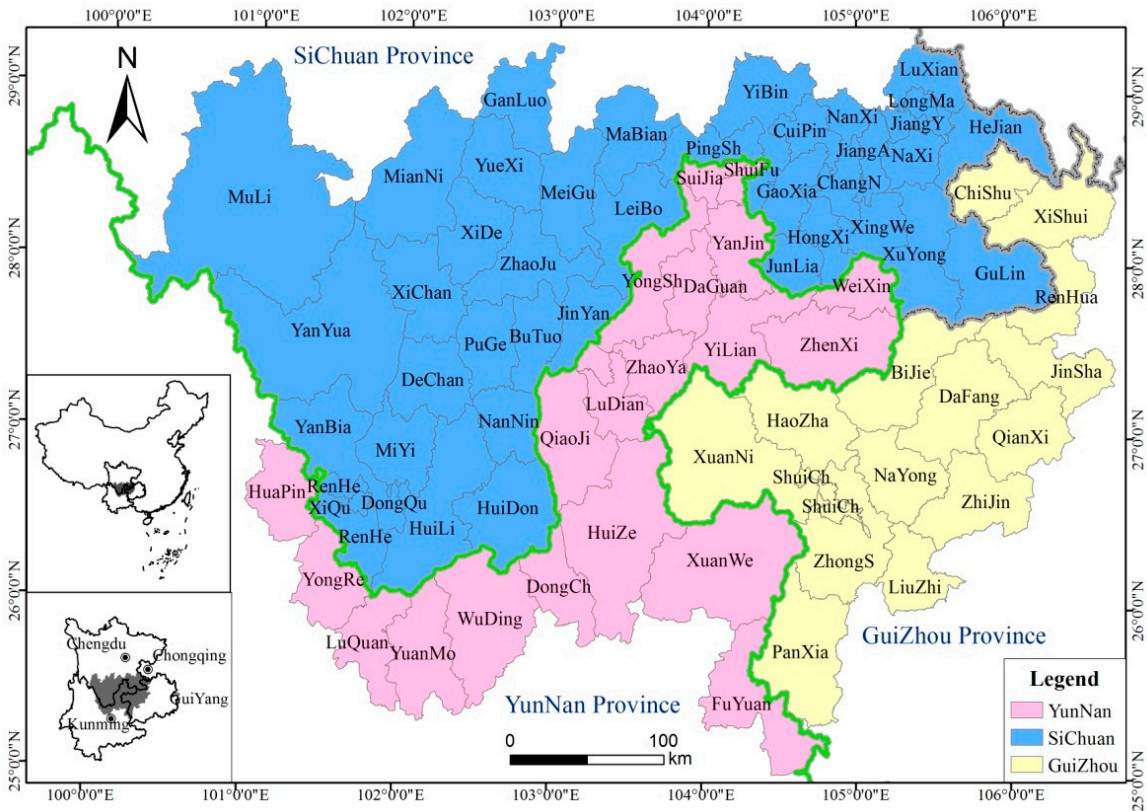

**Figure 1.** The location of the study area.

## 2.2. Method

### 2.2.1. ESDA Method

ESDA is considered to be a powerful tool for investigating regional spatial patterns and disparities, clusters, hotspots, and other forms of spatial heterogeneity [33–38]. We used ESDA to describe the spatial heterogeneity and the local spatial autocorrelation of economy and population gravity centres in order to explore the hot spot clustering. Global and local spatial autocorrelation analysis is commonly used for spatial dependence and heterogeneity research. Global spatial autocorrelation is a measure of overall clustering, which was measured here by the global Moran's *I*. The global Moran's *I* measures the degree of linear association between its value at one location and the spatially weighted average of the neighbouring value [39,40]:

$$I = \frac{\sum_{i=1}^{i=n} \sum_{j=1}^{j=n} W_{ij} (X_i - \overline{X})(X_j - \overline{X})}{S^2 \times \sum_{i=1}^{i=n} \sum_{j=1}^{j=n} W_{ij}} \tag{1}$$

$$S^2 = \frac{1}{n} \sum_{i=1}^{n} (X_i - \overline{X}) \tag{2}$$

where *n* is the number of counties in this study; $X_i$ and $X_j$ are the observations of regions *i* and *j*, respectively; $\overline{X}$ is the mean of *X*; and $W_{ij}$ is the element of the spatial weight matrix *W*. If regions *i* and *j* are adjacent to each other, then $W_{ij} = 1$, otherwise $W_{ij} = 0$.

Local spatial autocorrelation is used to assess the local structure of spatial autocorrelation by the local indicators of spatial association (LISA) [39,40].

$$I' = \frac{(X_i - \overline{X})}{S^2} \sum_{j=1}^{n} W_{ij} (X_j - \overline{X}) = Z_i' \sum_{j=1}^{n} W_{ij} Z_j', \tag{3}$$

where $Z'_i$ and $Z'_j$ are the standardised observations; and $W_{ij}$ is the element of the spatial weight matrix $W$.

The values of Moran's $I$ range from 1 to $-1$, where $I = 1$ suggests a perfect positive spatial autocorrelation (where high values or low values show a spatial cluster); $I = -1$ suggests perfect negative spatial autocorrelation (a checker board pattern); and $I = 0$ suggests perfect spatial randomness.

### 2.2.2. Gravity Centre Model

Economic and population gravity centres are the force point of the economic and population sub-vector in each region [13]. The gravity centre model was used to obtain the geographic coordinates T ($x_i$, $y_i$) of the economic and population indicators of each county in the JRSYG. The coordinates of the economy gravity centre are GE ($x_j$, $y_j$), and the population gravity centre coordinates are GP ($x_j$, $y_j$). The formulae are described as follows:

$$x_j = \frac{\sum(E_{ij} \cdot T(x_{ij}))}{\sum(E_{ij})}, \ y_j = \frac{\sum(E_{ij} \cdot T(y_{ij}))}{\sum(E_{ij})} \tag{4}$$

$$x_j = \frac{\sum(P_{ij} \cdot T(x_{ij}))}{\sum(P_{ij})}, \ y_j = \frac{\sum(P_{ij} \cdot T(y_{ij}))}{\sum(P_{ij})}, \tag{5}$$

where I is a county unit; *j* represents a particular year (*j* = 1995, 1996, . . . . . . , 2015); E is the GDP; and P is the population.

### 2.2.3. Migratory Distance Model

In order to effectively understand the dynamic process of the economy and population, the migratory distance was calculated using the migratory distance model. According to the direction and distance of the migration, we can determine the degree of imbalance and the migratory trend of the economy and population gravity centres.

Assuming that the gravity centre migratory distance of the first $k + 1$ year is d (relative to the first $k$ years), then the gravity centre migratory distance formula is determined as follows [14]:

$$d_{(k+1)-k} = c \cdot \sqrt{(x_{k+1} - x_k)^2 - (y_{k+1} - y_k)^2}, \tag{6}$$

where the constant $c$ is 111.111, which is the coefficient from the Earth's surface coordinate units (degrees) to a horizontal distance (km).

### 2.2.4. Spatial Coupling Model

As the distance between the economy gravity centre and population gravity centre represents the overlapping space (Superposition, $S^*$), the closer distance represents a higher overlap. In order to study the coordination of the economic and population gravity centres, we used the spatial coupling model to analyse the spatial overlap of the two gravity centres. The spatial overlap $S^*$ is calculated as follows [13]:

$$S* = d_{G_e \cdot G_p} = c \cdot \sqrt{(x_E - x_P)^2 - (y_E - y_P)^2}, \tag{7}$$

where $d_{GE \cdot GP}$ represents the distance between the population gravity centre and the economy gravity centre; $x_E$ is the economy gravity centre x coordinates of the corresponding year; $y_E$ is the economy gravity centre y coordinates of the corresponding year; $x_p$ is the population gravity centre x coordinates of the corresponding year; $y_p$ is the population gravity centre coordinates of the corresponding year; and the constant c is 111.111, which is the coefficient that converts the Earth's surface coordinate units (°) into a horizontal distance (km).

## 3. Results and Discussion

### 3.1. Spatial Heterogeneity

The ESDA method was applied to investigate the correlation and spatial heterogeneity of the economy and population in the JRSYG. The values of the global Moran's *I* for the economy and population during the period 1995–2015 were all positive (Figures 2 and 3), which indicated that there was positive clustering and confirmed the presence of a strong positive spatial autocorrelation among the counties. Counties with a high level of economic development were adjacent to each other, as were those with a low level of economic development. The same phenomenon also occurred with the population among the counties.

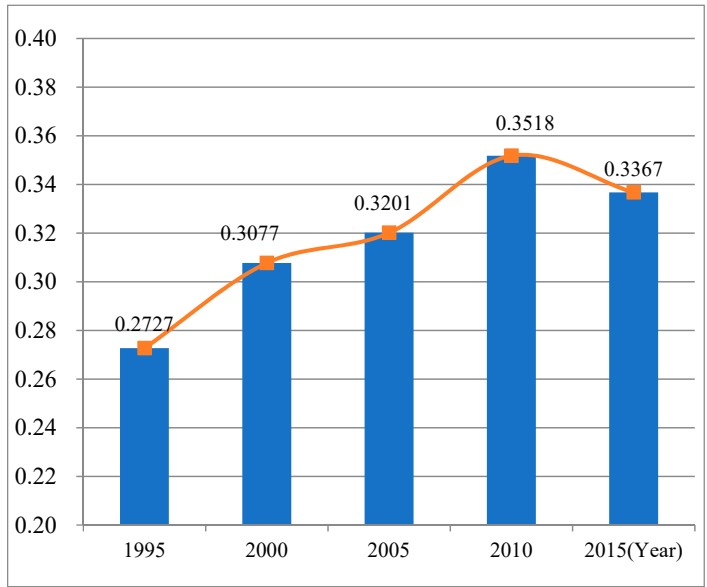

**Figure 2.** The global Moran's *I* index values of economy in the junction region of the Sichuan–Yunnan–Guizhou province (JRSYG).

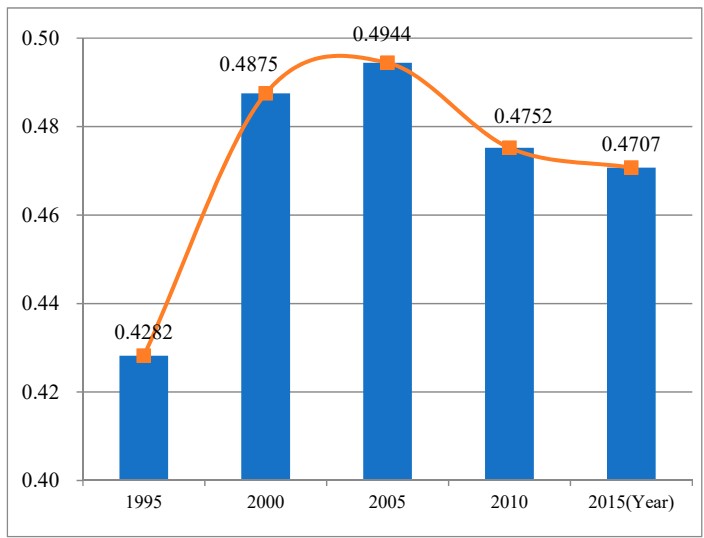

**Figure 3.** The global Moran's *I* index values of population in the JRSYG.

During the study period, the global Moran's *I* for the economy ranged from 0.2727 to 0.3518, which suggests that the distribution of the economy is a nature clustering. The global Moran's *I* for the population ranged from 0.4282 to 0.4948, which suggests that the distribution of the population

is also a nature clustering. It also indicates that the population had a stronger agglomeration than the economy. On average, the global Moran's *I* for the economy increased before 2010 and the global Moran's *I* for population increased before 2005. This also indicates that the degrees of economic and population polarization weakened from 2010 to 2015 and 2005 to 2015.

From the Moran scatter plot of the economy (Figure 4), we can see that the scatter plots were mainly distributed in quadrant III, with an L–L (a region with a low value surrounded by regions with low values) gathering. There were only a few distributed in quadrant I, with an H–H (a region with a high value surrounded by regions with high values) gathering. With the passage of time, the scatter plots distributed in quadrant I tended to increase. This not only shows that the agglomeration and driving effects of the economy became stronger, but also indicates that there was an increasing number of counties with a high level of economic development.

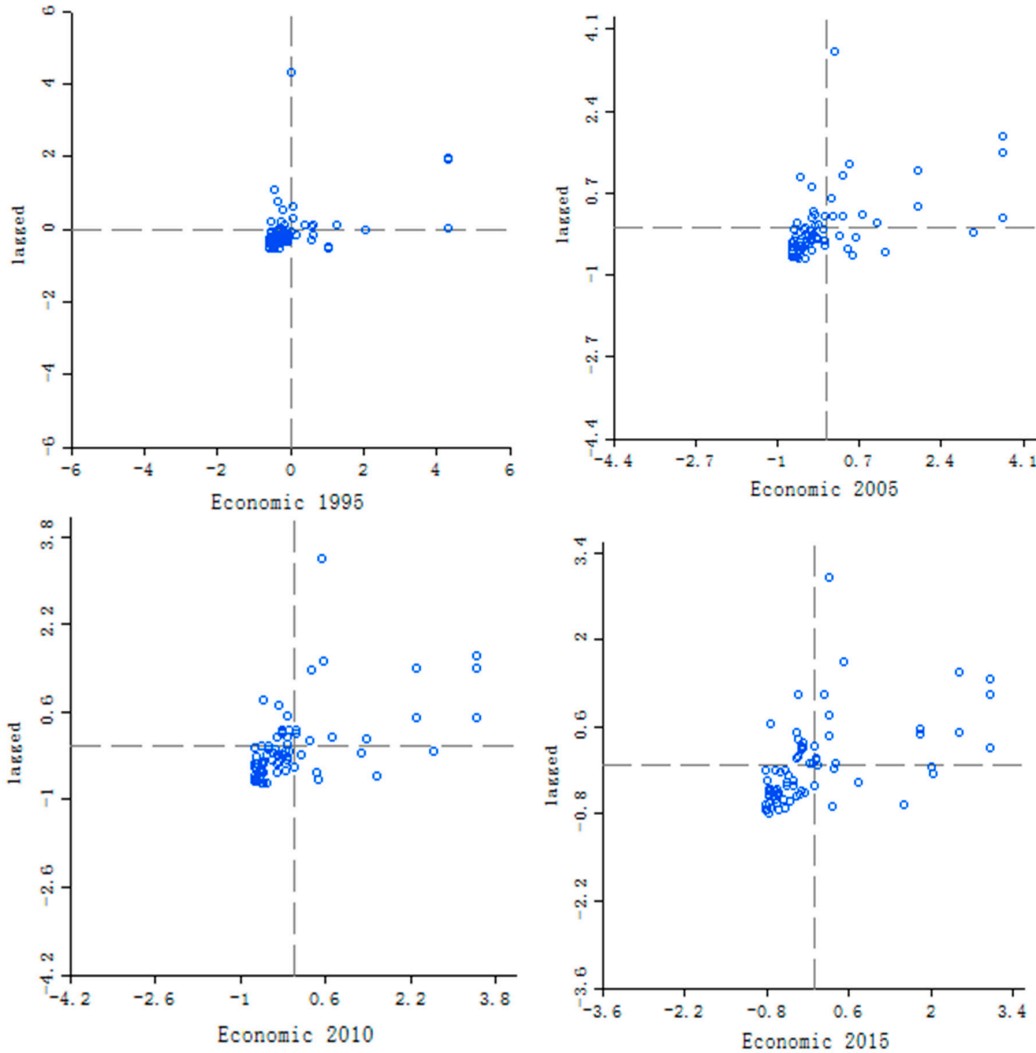

**Figure 4.** Moran scatter plot of the economy in the JRSYG during the period 1995–2015.

As shown in the Moran scatter plot of the population (Figure 5), the scatter plots are mainly distributed in quadrant III with an L–L gathering and in quadrant I with an H–H gathering. Only a few scatter plots are distributed in quadrant II with an L–H (a region with a low value surrounded by regions with high values) gathering and in quadrant IV with an H–L (a region with a high value surrounded by regions with low values) gathering, which means that there were only small disparities between regions with a negative spatial autocorrelation (H–L, L–H).

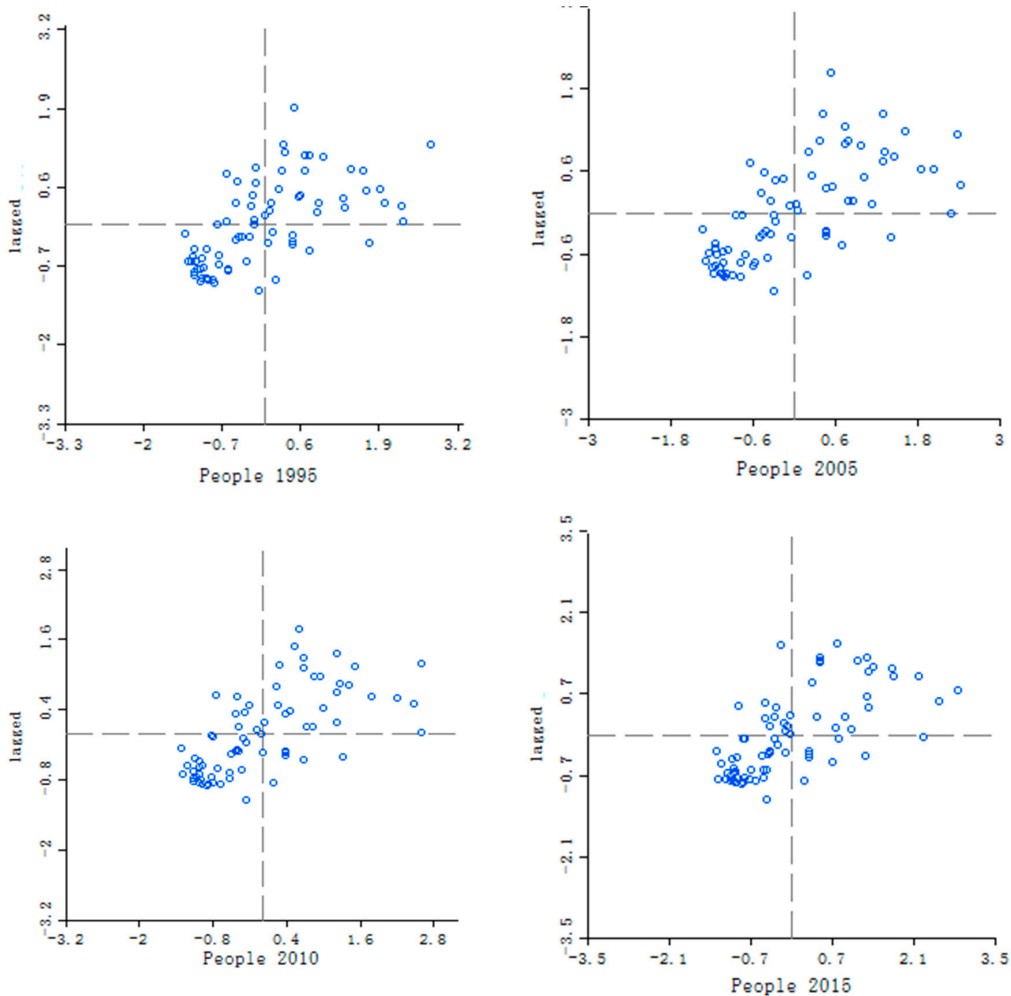

**Figure 5.** Moran scatter plot of the population in the JRSYG during the period 1995–2015.

### 3.2. Dynamic Evolution Trends

In order to effectively understand the dynamic processes of the economy and population, the economic and population gravity centres, GE ($x_j$, $y_j$) and GP ($x_j$, $y_j$), were calculated using the gravity centre model and migratory distance model, while the migratory distance was mapped using Sigmaplot 10.0.

The GE was 27.5406° N, 103.9975° E in 1995 and 27.5537° N, 104.2840° E in 2015. According to the trajectory of movement, the migratory distance of the GE was 123.82 km over the past 20 years (Figure 6). The migratory distance range of the GE was larger than that of the GP and was mainly concentrated on the border area between Zhenxiong and Yiliang Counties. Relative to the GE, the GP was always distributed in the eastward area, and their high-frequency areas did not intersect (Figure 6). The GP, which showed a clockwise spiral development trend, was 104.4288° E, 27.5162° N in 1995 and 104.4264° E, 27.5097° N in 2015. The GP moved 14.46 km in the past 20 years (Figure 6). The moving range of the GP was limited to the region between Zhenxiong County and Yiliang County in Yunnan Province. According to the dynamic evolution track of the two gravity centres (Figure 6), they are approaching each other and the regional disparities are decreasing.

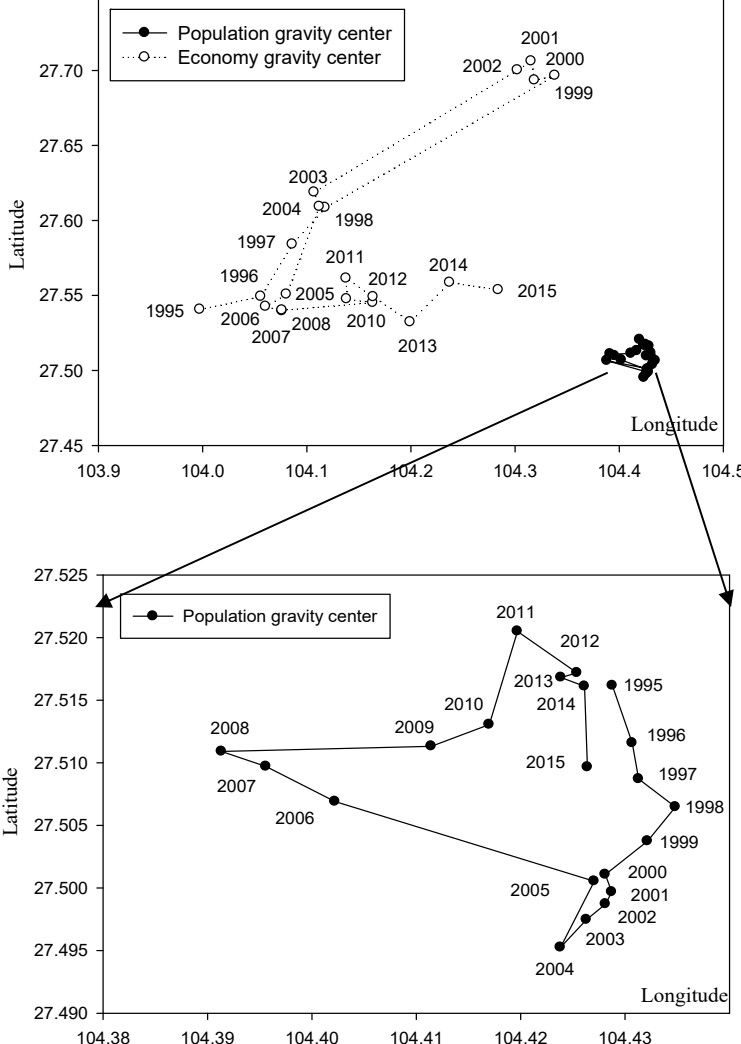

**Figure 6.** The dynamic evolution track of the economy gravity centre (GE) and the population gravity centre (GP) in the JRSYG during the period 1995–2015.

### 3.3. Spatial Coupling

The *S\** (the distance between the GE and GP) was calculated using the spatial coupling model of the gravity centre. For the *S\** representing the overlapping space, a closer coupling had a higher value. The greatest distance between the two gravity centres was 48.00 km, and the spatial coupling was the smallest in 1995. The closest distance between them was 16.56 km, and the spatial coupling was the largest in 2015 (Figure 7). The equilibrium point of the two gravity centres moved constantly, following an inverted U-shaped curve in the past few years. In terms of spatial overlap, the GE and GP gradually strengthened, indicating that their spatial coupling increased every year (Figure 7).

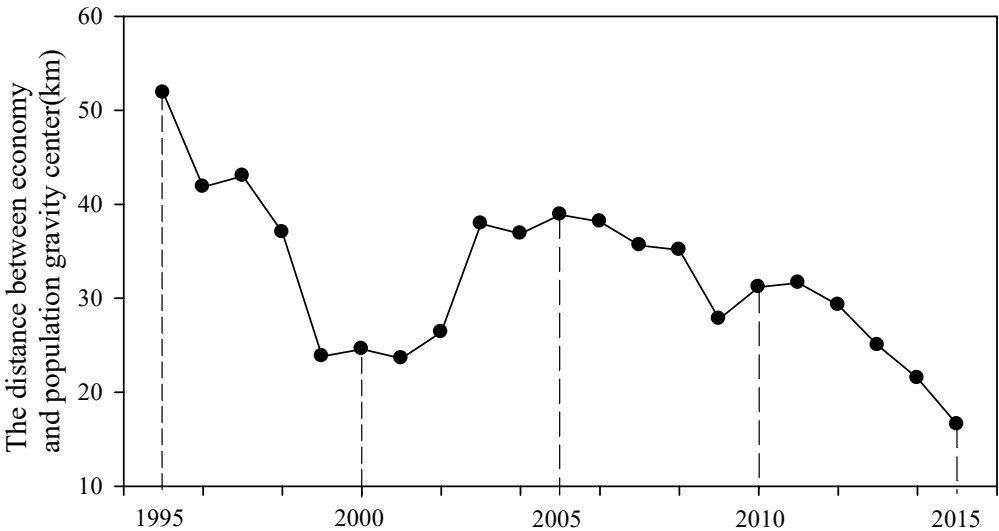

**Figure 7.** The coupling curves of the GE and GP in the JRSYG.

### 3.4. Hotspots for Economic Development

In order to provide more convincing evidence for the policy recommendations, LISA was used to assess the local spatial autocorrelation in order to explore the hotspot clustering.

It can be seen from Figure 8 that the proportion of L–L areas was relatively large, indicating a low level of economic development in the respective area and the surrounding counties. There was only one H–H area in 1995, which was Panzhihua city. After some time, the H–H areas were mainly located in Panzhihua city and Luzhou city. The H–L areas were distributed in Xichang City, Zhaoyang District, and Zhongshan District in 2015, which indicates that the levels of economic development were high, while that of the surrounding counties was low. From the LISA gathering map (Figure 8), a hotspot and sub-hotspot could be identified. Panzhihua city and Luzhou city were hotspots, and Xichang City and Zhaoyang District were the sub-hotspots.

According to the above analysis, a point–axis spatial development pattern (PAP) was suggested for the development of the JRSYG in the future (Figure 9). The PAP consists of five points, two axes, and two slices. The five points can be split up into two slices by the Jinshajiang River axis. The two slices are the core of Luzhou–Zhaoyang–Zhongshan and Panzhihua–Xichang. The two axes are composed of the main traffic arteries and the river. The traffic artery axes are the connecting lines, such as the national and provincial highways, railways, and other main traffic arteries. The river axis is the dividing line.

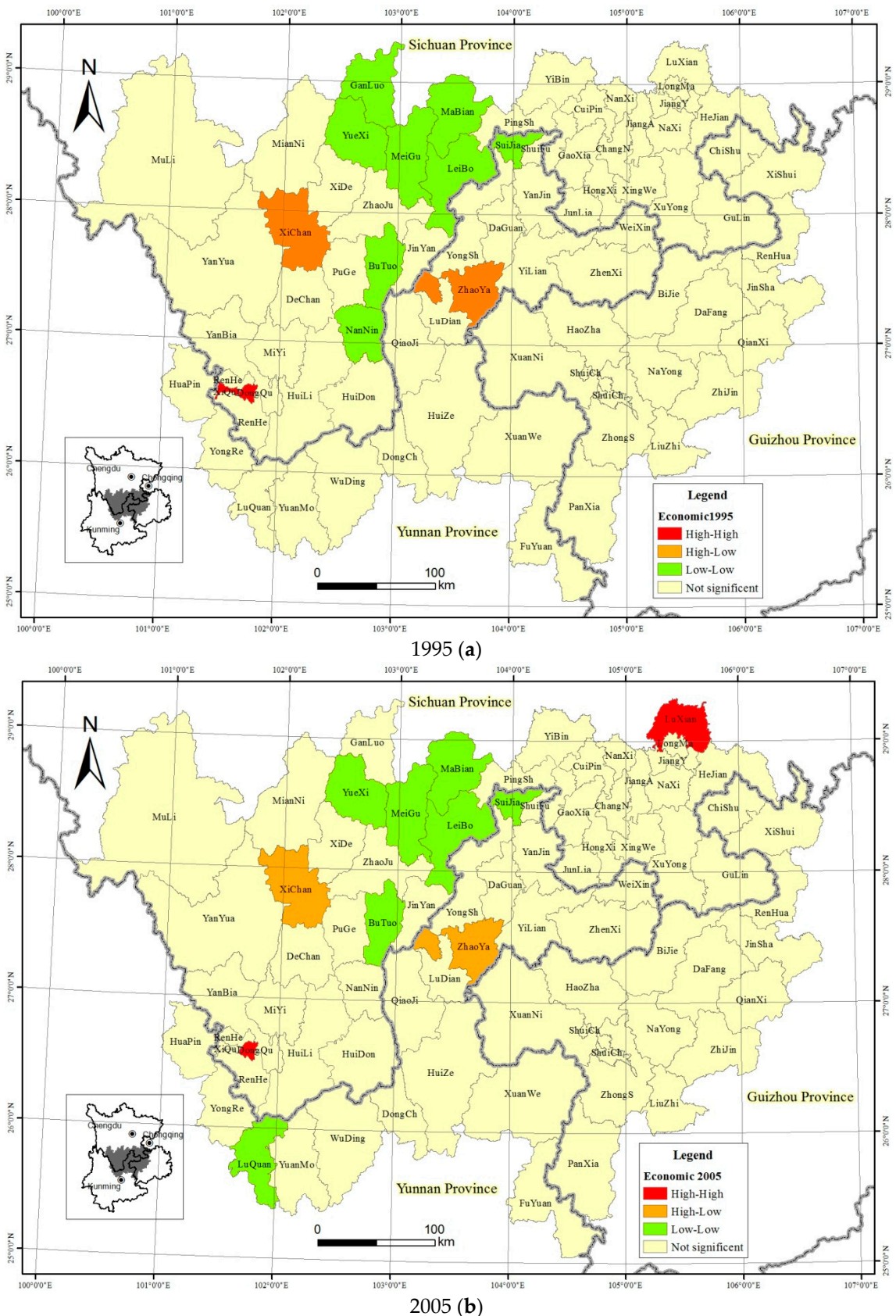

1995 (**a**)

2005 (**b**)

**Figure 8.** *Cont.*

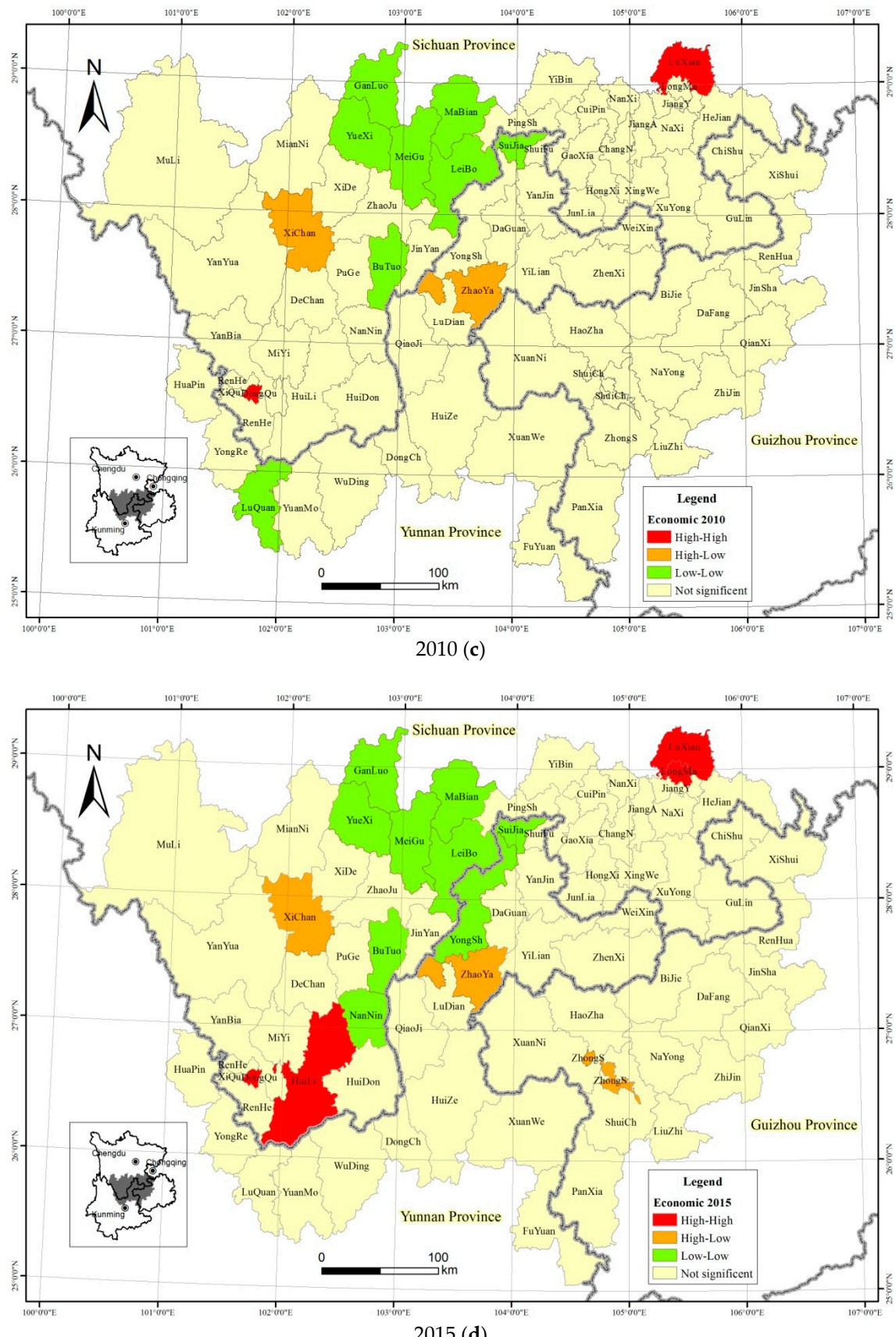

2010 (**c**)

2015 (**d**)

**Figure 8.** LISA gathering map of the GE in the JRSYG (5% significance level).

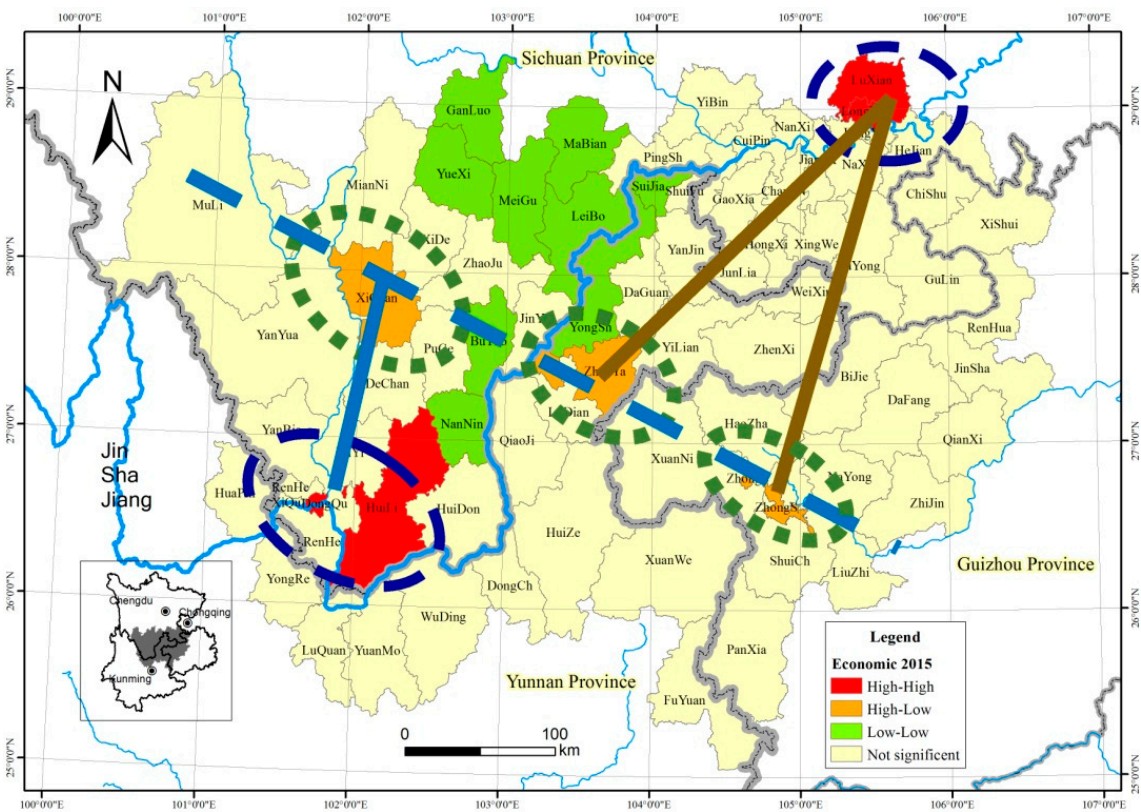

**Figure 9.** The point–axis spatial development pattern (PAP) in the JRSYG.

## 4. Discussion

Compared with previous studies [40–42], we not only determined the spatial heterogeneity and coupling of the economy and the population over the past 20 years, but also visually illustrated the PAP in the JRSYG. The growth hot poles can promote provincial cooperation, with further advantages gained from policy and infrastructure support. Therefore, this study may provide more convincing evidence for policy recommendations.

(1) The global Moran's *I* showed an inverted U-shaped curve (Figures 2 and 3), and its value nearly peaked in recent years, indicating that the spatial clustering of the economy and population is becoming stronger. From Figure 7, we can see that the spatial overlapping of the GE and GP gradually increased, indicating that their spatial coupling increased over time. Certain Chinese national policies facilitated this clustering process of the economy and population, such as the western development strategy, which matched the relevant project construction and investment, and the reform of China's household registration policy and household registration management.

The scatter plots of the population were mainly distributed in quadrant III and quadrant I (Figure 5). The scatter plots of the economy were mainly distributed in quadrant III (Figure 4) and the major type of spatial distribution was an L–L cluster, suggesting that the overall economic development was at a low level balanced state [41]. On the one hand, this shows that the spatial clustering of the county is relatively stable, and that it is difficult for a single county to get rid of the original cluster. On the other hand, it shows that while the spatial heterogeneity slowly increased, the clustering still dominated. With the passage of time, the agglomeration and driving effects of the economy become stronger, and the scatter plots of the economy that are distributed in quadrant I tend to increase.

(2) Over the past 20 years, the migratory range of the GE showed strong volatility but gradually became stable in recent years (Figure 6). From 1995 to 2005, the GE was mainly in the northwestern area. In contrast, it gradually shifted to the east from 2005 to 2015. With the further increase in intensity of the economic reform, the competition between economic zones—such as the Southern Sichuan Province

Economic Zone and the Panxi Economic Zone—caused a continuous fluctuation of the economic barycentre. The GE moved farther than the GP. The total distance that the GE moved was 123.82 km, while the GP moved 14.46 km. The difference in economic development was greater than the difference in population distribution. Compared with the economic barycentre, the migratory range of the GP was relatively small. From 2005 to 2015 the migration path of the GP became more obvious compared to that from 1995 to 2005. From 2005 to 2015, the GE gradually moved towards the GP. The spatial coupling states of the two barycentres show an inverted U-shaped cyclical change (Figure 7). Both of them are separated in the spatial distribution, but their spatial overlapping gradually increased, indicating that their spatial coupling increased over time. The Chinese government implemented the western development strategy and matched the relevant project construction and investment in 2000, which led to the economy gravity centre moving to the mainland [13]. On the other hand, population movement was restricted in the early 1990s, and as China's household registration policy and household registration management were relaxed, the population movement increased [13,43]. Therefore, the overall effect has been for the regional disparity in the past ten years to narrow, but it still shows the essence of unbalanced development [13].

(3) According to the LISA gathering map (Figure 8), the economic development level presents a polarisation pattern with a southwest growth pole and a northeast growth pole. The economic hotspots and sub-hotspots in the JRSYG mainly revolved around five centres, and present a core–edge echelon-structured pattern. The hotspots were Panzhihua and Luzhou in Sichuan province. The sub-hotspots were Xichang in Sichuan province, Zhaoyang in Yunan province, and Zhongshan in Guizhou Province. The H–H areas were mainly located in the capital of the city and state, which has advantages and development opportunities in areas such as politics, economy, transportation, science and technology, education, etc. The L–L areas were mainly distributed in Liangshan Mountain, which has a weaker economic foundation as well as fewer resources and less transportation infrastructure.

The economy and the population have a symbiotic relationship and influence each other. The regions with an advanced economy were also the agglomerated regions with a high-level of coupling coordination of urbanization. Accordingly, it is important to enhance the spillover effects and driving effects of the High–High cities on neighbouring cities [44], and reduce the impact of the administrative boundaries [41] between the Sichuan, Yunnan, and Guizhou provinces, in order to realise regional economic integration. Disadvantageous geographic and environmental conditions restrict regional development [42]. More reasonable development strategies are necessary for these regions in order to promote the coordinated development of the population and economy [45,46]. Apart from developing transportation and other infrastructure and strengthening their industrial development, promoting the growth poles is a better strategy to promote provincial cooperation and achieve a high level of balanced development in the future. The primary task of achieving coupling coordination for these Low–Low areas may still involve the development of the economy [47]. These five cities as the growth poles would drive the economic and social development of the surrounding L–L areas.

## 5. Conclusions

This article analysed the spatial heterogeneity and coupling of the economy and population in the JRSYG during the period 1995–2015. The results showed the following:

(1) According to the global Moran's *I* for the economy and the population, there was positive clustering and we confirmed the presence of strong positive spatial autocorrelation among the counties. The distribution of the economy and population was a nature clustering.

(2) According to the Moran scatter plot, the scatter plots of the economy and the population were mainly distributed in quadrant III, with an L–L gathering. With the passage of time, the scatter plots of the economy distributed in quadrant I tended to increase, showing that the agglomeration and driving effects of the economy increased.

(3)   Due to the dynamic evolution process of the economy and the population, the migratory distance of the GE was larger than that of the GP, but they were mainly concentrated in the border area between Zhenxiong and Yiliang Counties in Yunnan Province. The GE and GP are approaching each other, and their regional disparities are decreasing. The equilibrium points followed an inverted U-shaped curve every year. In terms of spatial overlap, the spatial coupling of the GE and GP increased year by year. The spatial relationship between them was more coordinated and the degree of balanced regional development continuously improved.

(4)   The point–axis spatial development pattern was presented, with two economic hotspots and three sub-hotspots in the JRSYG. If further advantages from policy and infrastructure support are obtained, the growth poles can promote provincial cooperation, alleviate regional differences, and achieve a high level of balanced development in the future, which could increase the competitiveness of the west and promote regional sustainable development.

**Author Contributions:** Data curation, S.L. and Y.L.; Formal analysis, H.Y.; Methodology, Y.L.; Supervision, S.L.; Writing—original draft, H.Y. and Y.L.; Writing—review and editing, H.Y. All authors have read and approved the final manuscript.

**Funding:** The research was funded by the National Natural Science Foundation of China (Grant No. 41671529), National Science and Technology Major Project for Water Pollution Control and Treatment (Grant No. 2017ZX07101001), Key Research and Development Projects in Sichuan Province (Grant No. 2019YFS0467), Hundred Young Talents Program of the Institute of Mountain Hazards and Environment (Grant No. SDSQB-2015-01), Monitoring and Warning Program for Resources and Environment Carrying Capability in Sichuan Province (Grant No. ZXGH201709), Knowledge Innovation Program of Chinese Academy of Sciences (Grant No. KZCX2-YW-333).

**Acknowledgments:** We thank the academic editors and anonymous reviewers for their kind suggestions and valuable comments.

**Conflicts of Interest:** The authors declare no conflict of interest.

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
