# Peer review of "Spatial Heterogeneity and Coupling of Economy and Population Gravity Centres in the Hengduan Mountains"

_sustainability, doi:10.3390/su11061508_

Round 1
Reviewer 1 Report
Dear authors.
The presented paper tries on analyzing the spatial distribution of economic and population centers in a Chinese region. This objective seems quite interesting to me, although I have some serious and important doubts about the acceptance of the article.
1.- I do not find the topic of the article directly related to those of the journal. Although the study of the spatial distribution of economic and population centers may help to design measures for the search for sustainability, this relationship is not sufficiently clear in the article.
2.- The paper presents a very poor English language level. Grammar, language and style MUST BE EDITED in an extensive review.
3.- The methodological section must be completed, explaining in a better way all the methods and indicators used, why they have been chosen, and what are the expected results when applying them.
4.- The discussion section would improve if it includes a broad economic and population analysis. The paper focuses on the Hengduan Mountains region, but when analyzing economy or population movements, national variables must be considered. In example, are there any Chinese national policy that explains the observed changes in the indicators proposed? Has the performance of the Chinese economy had anything to do with the evolution of indicators in recent years? What is the reason for the change in the evolution of any of the indicators?
Some other details:
Line 21: mistake with “population gravity centre”.
Line 35-36: this could be rewritten as “The economic and population gravity centres”.
Line 52: do not understand.
Line 54: explain again what does RSYG stands for.
Figure 1: a national map of China would help in understanding where the study area is located.
Equation 4: what does “E” stands for?
Line 199: what does “LISA” stands for?
Line 215: mistake with “previous study”, correct to “previous studies”.
Author Response
Dear Reviewer,
Thank you for your suggestion. The revision explanation about the manuscript sustainability-452687 is in the attachment. Grammar, language and style have been edited by MDPI (English editing ID: English-8215). We look forward to hearing from you soon.
Kind regards,
Authors

Reviewer 2 Report
This paper seems to very suitable to Sustainability because the authors communicate the results of highly-important research. The paper is generally well-written. Of course, certain improvements are necessary before acceptance.
1) What does mean 'junction region'? This term should be well-explained in the text, and I tend to recommend its deletion from the title to make the latter more clear.
2) The abstract should be less technical. No-meaning phrases like 'People pay more and more attention to this region.' should be totally avoided.
3) The abstract and the introduction should state how the scope of this paper is relevant to the issue of sustainable development.
4) Please, explain in the discussion why 'the spatial clustering of economy and population is becoming strong'.
5) I strongly encourage the authors to compare the results of their study with the results of some other, more or less similar studies undertaken in the other regions of the world.
6) In the conclusions, please, tell less about the methods and metrics. You need to present the only principal research outcomes there.
7) Please, explain all abbreviations when these are given for the first time in the text.
8) The number of literature sources by non-Chinese authors is insufficient in this paper, and it MUST be increased substantially. This is so because the paper is submitted to the world-class journal with the international readership – if so, the international research experience should be referred.
Author Response
Dear Reviewer,
Thank you for your suggestion. The revision explanation about the manuscript sustainability-452687 is in the attachment.Grammar, language and style have been edited by MDPI (English editing ID: English-8215). We look forward to hearing from you soon.
Kind regards,
Authors

Round 2
Reviewer 1 Report
Dear authors.
Congratulations
on the new version of your article, which has undoubtedly gained in
quality. I appreciate the effort made to incorporate the requested
changes and respond to our suggestions. The resulting article may be
accepted for publication in Sustainability.
If possible, I should suggest a better analysis of the obtained results as a way to enhance a bit more the article.